# A Real-Time Lightweight Behavior Recognition Model for Multiple Dairy Goats

**DOI:** 10.3390/ani14243667

**Published:** 2024-12-19

**Authors:** Xiaobo Wang, Yufan Hu, Meili Wang, Mei Li, Wenxiao Zhao, Rui Mao

**Affiliations:** 1College of Information Engineering, Northwest A&F University, Yangling 712100, China; xiaobowang@nwafu.edu.cn (X.W.); huyufan@nwafu.edu.cn (Y.H.); wml@nwsuaf.edu.cn (M.W.); limei@nwsuaf.edu.cn (M.L.); 2Shaanxi Engineering Research Center of Agriculture Information Intelligent Perception and Analysis, Yangling 712100, China; 3College of Animal Science and Technology, Northwest A&F University, Yangling 712100, China; zhaowenxiao@nwafu.edu.cn

**Keywords:** dairy goat, deep learning, behavior recognition, abnormal behaviors, GSCW-YOLO

## Abstract

Accurate real-time recognition of abnormal behavior in dairy goats facilitates timely intervention, thereby improving both health status and farming efficiency. This study presents GSCW-YOLO, a real-time behavior recognition model designed specifically for dairy goats. To meet the dual demands of real-time processing and high accuracy in recognizing multiple abnormal behaviors, the proposed model can accurately recognize six common behaviors (standing, lying, eating, drinking, scratching, and grooming) and four abnormal behaviors (limping, attacking, death, and gnawing). Compared to existing popular methods, GSCW-YOLO achieves superior accuracy and speed, offering a novel perspective for welfare assessment in dairy goats.

## 1. Introduction

With the advancement of large-scale and intensive modern livestock husbandry, the health and physiological conditions of livestock have become a major issue of increasing concern [1,2]. There is an inherent connection between the livestock behavior and their health and welfare [3,4]. Environmental changes or disease outbreaks often result in deviations from normal behavior patterns [5]. Additionally, consumer demand for animal welfare in farming practices is rising, with a preference for products from sources that adhere to higher welfare standards [6,7]. Consequently, intelligent analysis and precise recognition of livestock behavior are essential for maintaining livestock health and welfare [8].

The current standard method for recognizing abnormal livestock behavior is manual observation, which is time-consuming, inefficient, highly subjective, and unsuitable for the needs of intensive production [9,10]. Fortunately, advancements in sensor and computer vision technologies have led to a transition towards automated livestock behavior recognition methods [11,12]. These methods are categorized into contact and non-contact approaches [13].

Contact automatic recognition methods involve the use of wearable devices equipped with sensors for behavior recognition [14]. Shen et al. [15] used a triaxial accelerometer to extract time-domain and frequency-domain features, successfully identifying feeding and rumination behaviors in cattle using the k-nearest neighbor algorithm, achieving accuracies of 92.8% and 93.7%, respectively. Kleanthus et al. [16] achieved a recognition accuracy of 98.6% for grazing, active, and inactive behaviors in sheep by combining accelerometers and machine learning algorithms. Nevertheless, attaching or implanting sensors on livestock for contact-based detection may lead to data collection influenced by vibrations or noise, potentially causing harm or stress reactions in livestock and adversely affecting their welfare [17,18].

Non-contact methods typically employ computer vision and deep learning techniques to recognize livestock behavior [19]. Yin et al. [20] employed EfficientNet for feature extraction and a BiLSTM module to aggregate video frames in a time series, recognizing cow behaviors such as lying, standing, walking, drinking, and eating. The recognition accuracy of the model reached 97.8%. Ji et al. utilized an improved ResNet model to identify aggressive behavior in pigs, achieving a recognition accuracy of 95.7% [21]. Non-contact methods ensure undisturbed natural behavior observation of livestock, significantly reducing the stress or discomfort associated with data collection. These methods provide real-time, continuous, and remote recognition capabilities [22].

Understanding abnormal behaviors in dairy goats is vital for assessing their welfare. Dairy goats exhibiting lameness may indicate leg injuries or interdigital dermatitis infections [23]. Due to the damp environments that foster bacterial growth in their habitat, goats may bite their joints [24]. Additionally, aggressive behaviors in dairy goats can lead to injuries, stress, and severe infections. Farmers require information on these potential problem behaviors to assess and implement appropriate management strategies in a timely manner [25,26].

Advancements in non-contact behavior recognition methods have enabled real-time monitoring and analysis of sheep behavior using images and videos. Gu et al. [27] employed a two-stage recognition method to optimize the detector and achieved classification accuracies exceeding 94% for all behaviors using VGG. The YOLO series models are simple yet precise one-stage recognition models, and many studies have demonstrated their effectiveness in sheep behavior recognition. Jiang et al. [28] utilized the YOLOv4 model to recognize behaviors in goats, including eating, drinking, and being active or inactive, with average recognition accuracies of 97.9%, 98.3%, 96.9% and 96.9%, respectively. Cheng [13] et al. proposed a YOLOv5-based behavior recognition model for sheep, collecting data from four distinct sheep pens, which facilitated the recognition of behaviors such as standing, lying, eating, and drinking. The model achieved an overall recognition precision of 96.7% and a recall of 96.5%.

However, existing research exhibits certain limitations. Most studies prioritize the identification of individual livestock behaviors, predominantly concentrating on typical behavior categories and scarcely addressing abnormal behaviors [29]. Regarding the application scenarios pertinent to this study, the complex backgrounds inherent in real farming environments frequently diminish the model’s ability to concentrate on the distinctive aspects of sheep behavior. Additionally, stationary camera positions limit the model’s ability to precisely recognize behaviors of small targets within intricate environments.

To address these challenges, this study developed the GSCW-YOLO model, designed to accurately recognize multiple dairy goats’ behaviors. It aims to mitigate complex background interference, enhance the model’s capabilities in feature extraction and fusion, and assist farmers in timely detecting abnormal dairy goat behaviors to reduce losses. This model enhances animal welfare through improved health monitoring, supports sustainable livestock farming by optimizing resource use, and promotes ethical farming practices. Specifically, to improve the model’s multi-scale feature fusion capability, a small-target detection layer was added, transforming the existing two-layer Feature Pyramid Network (FPN) and Path Aggregation Network (PAN) configuration into a three-layer structure. Moreover, the study replaced the original nearest-neighbor upsampling operator in the FPN with a Content-Aware Reassembly of Features (CARAFE) operator and introduced a Gaussian Context Transformer (GCT) in the PAN’s final layer. These modifications improved the effective use of contextual information, minimized background noise interference, and prioritized detecting dairy goats. Lastly, the study proposed a dataset, GoatABRD, consisting of 9213 images capturing multi-scale goat behaviors in complex settings, including both abnormal (limping, attacking, death, and gnawing) and common (standing, lying, eating, drinking, scratching, and grooming) behaviors, thus broadening the scope of behavioral categories in dairy goat recognition.

## 2. Materials and Methods

### 2.1. Dataset Construction

#### Data Source

This study focused on dairy goats from the Guanzhong Plain region in central Shaanxi Province. Systematic data collection occurred from 6 October 2022 to 15 November 2023 at Onik Dairy Goats Farm in Liujizhen, Fuping County, Shaanxi Province, China, daily from 7:00 AM to 10:00 PM. The objective was to capture video recordings under varying lighting conditions to ensure a wide representation of lighting variations within the dataset, thereby improving the method’s adaptability to different lighting situations. Figure 1 displays selected shooting scenes. Figure 2a details the outdoor camera installation sites and the dairy goat shed’s design, while Figure 2b illustrates the indoor camera setup, including the feeding and drinking areas. The cameras, the Hikvision DS-7808N-SN model, were mounted around 3.6 m above the ground at a roughly 30-degree angle. Figure 2a shows a shed with sixteen dairy goats separated by an aisle, with eight dairy goats on each side, and Figure 2b highlights a single dairy goat shed with its feeding zone marked in green, a water trough, and an outdoor area. Each dairy goat shed typically accommodated 15 to 18 dairy goats. The goats were kept in their natural environment with unrestricted movement throughout the study, ensuring their welfare and the authenticity of the collected data.

This study analyzed ten behaviors observed in dairy goats, including limping, attacking, death, gnawing, standing, lying, eating, drinking, scratching, and grooming. Detailed definitions of these behaviors are delineated in Table 1.

### 2.2. Data Preprocessing

The data preprocessing for this study encompasses four stages: image acquisition, filtering, annotation, and augmentation. Initially, image acquisition involved capturing videos of ten distinctive dairy goat behaviors using specialized recording equipment and storing them on a network hard drive. A Python 3.7 script was then created to extract every tenth frame from these videos, saving the frames in .jpg format to avoid data redundancy caused by the high similarity of consecutive frames. In the filtering stage, images were scrutinized to discard those significantly occluding dairy goat behaviors, leaving 9213 viable images. To maintain dataset integrity and precision, manual annotation was conducted under the guidance of professional livestock experts using LabelImg, a graphical image annotation tool, where various behaviors were systematically categorized and tagged, with annotation data preserved in YOLO format.

The dataset was enhanced using various techniques, including image stitching, mirroring, cropping, random rotation, and HSV color tone adjustment. These methods were specifically employed to address the low occurrence frequency of abnormal behaviors and to generate additional samples representing these rare behaviors, thereby enhancing the dataset’s diversity and improving the model’s robustness. Furthermore, to more accurately mirror real-world conditions of small, distant targets, image resolution was adjusted to 224 × 224 pixels. This strategy aimed at augmenting the presence of small targets within the dataset, reflecting the reality of varying distances in field observations. Additionally, recognizing the diversity of behaviors among dairy goats in single images, the GoatABRD dataset labels each behavior instance across 9213 images, as detailed in Table 2. The dataset was partitioned into training, validation, and test sets in a 7:2:1 ratio, using the training set for model training and the validation and test sets for model evaluation.

### 2.3. GSCW-YOLO Behavior Recognition Model

This study introduces an optimized model, GSCW-YOLO, aimed at crafting a lightweight framework capable of achieving fast recognition speeds while addressing the challenges of performance degradation encountered in tasks involving lower image resolution or smaller objects. The GSCW-YOLO model, depicted in Figure 3, features a tripartite architecture consisting of a backbone, neck, and head components. The data frame is transmitted through the backbone network, which extracts fundamental spatial features for detailed representation. Subsequently, the neck module employs an enhanced FPN and PAN structure to fuse multi-scale features, thereby improving contextual awareness and reducing background noise. Finally, the head module utilizes the fused features for precise object detection.

The backbone’s primary role is key feature extraction, incorporating elements such as the Conv and C2f modules. The Conv module performs convolution and batch normalization utilizing the SiLU activation function on the input image, while C2f, pivotal for residual feature learning, achieves a balance between maintaining a lightweight structure and capturing extensive gradient flow details.

The neck segment plays a crucial role in merging features across various scales to create a unified feature pyramid. Its basic configuration includes the Feature Pyramid Network (FPN) [30] and the Path Aggregation Network (PAN) [31], enriched with the CARAFE [32] upsampling operator and GCT [33] to enhance context awareness and reduce the impact of background noise on feature extraction. This modification not only improves recognition accuracy but also significantly reduces the model’s parameter count. Consequently, the customary nearest neighbor interpolation in FPN has been replaced with CARAFE, integrating GCT into PAN for superior efficiency. To improve the model’s sensitivity towards smaller targets, an additional layer was added to the initial dual-layer setup in both FPN and PAN, as illustrated in Figure 3a, thereby enhancing multi-scale feature fusion capabilities.

The head component processes feature maps of various sizes to identify category and positional information of objects across different scales. The introduction of a high-resolution detection head, as shown in Figure 3a, improves accuracy in identifying smaller-scale targets. Furthermore, the detection head’s Bbox loss function CIoU was substituted with Wise-IoU, indicated in Figure 3d, to better accommodate positional variations in dairy goats and to expedite training convergence.

The CARAFE operator is essential in preserving fine details and contextual information within the feature map. The GCT module reassigns weights to mitigate the interference caused by complex backgrounds, thereby enabling the model to focus more effectively on goat behavior. The SOD module aids in capturing fine-grained details, significantly enhancing the detection of small targets. Furthermore, the Wise-IoU loss function enables the model to accurately capture transient interactions, leading to faster convergence and enhanced localization accuracy. Further details of these modules are provided below.

#### 2.3.1. Lightweight Upsampling Operator CARAFE

The CARAFE upsampling operator is a lightweight yet versatile system, comprising two core components: the kernel prediction module and the content-aware reassembly module, as represented in Figure 3b. The process begins with the kernel prediction module, which determines the necessary upsampling kernels before the content-aware reassembly module performs the upsampling, resulting in the output feature map.

Given a feature map χ with the dimensions of C×H×W and an upsampling factor σ, the kernel prediction module initiates the process with a 1 × 1 convolutional layer that compresses the feature channels from C to $C_{m}$, thereby reducing computational load. This is followed by content encoding and upsampling kernel prediction in the compressed feature map, which has the dimensions of $C_{m}\times H\times W$. A convolutional layer with a size of k×k is utilized to predict the upsampling kernel, taking $C_{m}$ input channels and producing $\sigma^{2}k_{up}^{2}$ output channels. The resulting tensor is reshaped from channel dimensions to spatial dimensions, creating an upsampling kernel with the dimensions of $k_{up}^{2}\times \sigma H\times \sigma W$, which is then normalized using softmax to ensure the sum of kernel values equals 1.

In the succeeding content-aware reassembly module, each feature point in the output feature map is linked back to a corresponding point in the input feature map. A region of size $k_{up}\times k_{up}$ is extracted around each feature point, and the dot product is computed with the corresponding upsampling kernel. Notably, different channels at identical positions utilize the same upsampling kernel, culminating in a new output feature map χ′ with the dimensions of C×σH×σW. This methodology facilitates content-aware upsampling while preserving spatial coherence and channel fidelity throughout the feature map.

The CARAFE operator is particularly adept at conserving intricate details and pertinent contextual information in feature maps, which is paramount for the nuanced recognition of dairy goat behaviors. This capability allows for the detailed capture of diverse backgrounds associated with specific actions or postures, thereby enhancing the model’s accuracy in behavior recognition, especially in dynamic and unpredictable contexts. The operator’s improved contextual comprehension markedly boosts the model’s precision in identifying behaviors.

#### 2.3.2. Gaussian Context Transformer (GCT)

The GCT introduces an innovative channel attention mechanism based on the negative correlation between global context features and attention activation values. This mechanism employs a Gaussian function to convert global attention into an attention map, accurately reflecting a predefined negative correlation. Illustrated in Figure 3c, the GCT’s architecture comprises three primary processes: Global Context Aggregation (GCA), Normalization, and Gaussian Context Excitation (GCE).

GCA utilizes global average pooling (GAP) to consolidate global context features across the spatial dimensions of each sample, producing channel descriptors that encapsulate global information of the feature map’s spatial aspects. For an input feature map χ, with the dimensions of C×H×W, where C, H, and W denote the channel count, height, and width, respectively, GAP processes this map into a global context vector z, which is then normalized and processed to derive attention weights.

Normalization ensures that z maintains a distribution with zero mean and unit variance across channels. The application of a Gaussian function, characterized by amplitude a, mean b (set to zero), and standard deviation c, facilitates the alignment with the assumed negative correlation by adjusting the excitation values g. Here, amplitude a regulates the excitation range within interval (0, 1], and standard deviation c dictates the variety within the channel attention map. Following Gaussian excitation, g represents the attention weights for each channel. These weights are applied element-wise to the original feature map χ, producing an enhanced, attention-focused feature map Y.

Unlike the original YOLOv8n model, which lacks a dedicated mechanism for weight distribution during feature extraction, leading to indiscriminate attention across background and behavior-specific details, the GCT employs the Gaussian function to allocate attention weights strategically. This selective weighting accentuates dairy goat behaviors while suppressing background noise, significantly improving behavioral recognition accuracy. The GCT’s efficient design also contributes to the development of a more streamlined and rapid recognition model, underlining its potential for enhancing precision in behavior analysis.

#### 2.3.3. Small Object Detection (SOD) Layer

Enhancements to the two-layer FPN and PAN within the Neck network include additional layers, evolving them into three-layered structures. The first layer of the PAN, characterized by a smaller downsampling factor, is preserved and augmented as an extra detection head. The specific structure of the SOD layer, detailed in Figure 3a and marked in red, comprises three FPN layers (FPN3), a single PAN layer (PAN1), and the newly incorporated detection head.

By retaining the initial PAN layer output with a reduced downsampling rate as the detection head, the SOD layer achieves finer resolution, facilitating the capture of intricate image details and effective small target detection. Conversely, the deeper detection head, denoted as p5, with its lower resolution and larger receptive field, is adept at encompassing broader areas, making it ideal for identifying larger targets. This fusion significantly enhances the model’s feature fusion ability, addressing the original YOLOv8n model’s limitation, where a larger downsampling factor limited the learning of small object features, thus improving the model’s ability to recognize small targets.

#### 2.3.4. Optimization of Loss Function

The Wise-IoU [34] loss function dynamically adjusts gradient weights corresponding to diverse behaviors, thereby augmenting the model’s adaptability to subtle behavioral fluctuations observed in dairy goats. It emphasizes recurring changes and interactions between behaviors, leading to improved accuracy in detecting abnormal behaviors in dairy goats. Wise-IoU’s structure is depicted in Figure 3d.

Inspired by the monotonic aggregation mechanism of focal loss [35] for cross-entropy, Zanjia et al. introduced a monotonic attention coefficient. This coefficient is designed to adaptively modulate the weights of anchor boxes, spotlighting significant ones while mitigating the negative influence of less effective anchor boxes on the gradient. Additionally, the inclusion of a mean as a normalization factor in Wise-IoU facilitates a quicker model convergence during advanced training phases. The formulas constituting Wise-IoU are as follows:(1)$$L_{IoU}=1-IoU=1-\frac{W_{i}H_{i}}{S_{u}}$$
(2)$$R_{IoU}=exp(\frac{{\left(x-x_{gt}\right)}^{2}+{\left(y-y_{gt}\right)}^{2}}{{\left({\left(w_{g}\right)}^{2}+{\left(h_{g}\right)}^{2}\right)}^{*}})$$
(3)$$L_{Wise-IoU}={\left(\frac{L_{IoU}^{*}}{\bar{L_{IoU}}}\right)}^{\gamma }R_{IoU}L_{IoU}={\left(\frac{L_{IoU}^{*}}{\bar{L_{IoU}}}\right)}^{\gamma }exp\left(\frac{{\left(x-x_{gt}\right)}^{2}+{\left(y-y_{gt}\right)}^{2}}{{\left(W_{g}^{2}+H_{g}^{2}\right)}^{*}}\right)(1-\frac{W_{i}H_{i}}{S_{u}}$$
where $L_{IoU}$ signifies the loss value of the intersection-over-union (IoU) ratio. The width and height of the predicted and real regions are denoted by $W_{i}$ and $H_{i}$ respectively, which are utilized to compute the area of the intersection between the two regions. The union area of predicted and actual regions is represented by $S_{u}$. The exponential function, signified as exp, is used to modulate the influence of the distance between two entities. The coordinates x and y are the center point of the predicted bounding box, while $x_{gt}$ and $y_{gt}$ denote the coordinates of the center point of the target bounding box. The width and height of minimum bounding rectangle of the target box, indicating the actual size of the target, are represented as $W_{g}$ and $H_{g}$ respectively. The superscripts ***** in ${\left(W_{g}^{2}+H_{g}^{2}\right)}^{*}$ denotes operations detached from the computational graph to prevent the gradient from obstructing convergence of $R_{IoU}.\bar{L_{IoU}}$ is the average loss value of the intersection ratio, while $L_{IoU}^{*}$ represents gradient gain, $L_{IoU}^{*}\in [0,1]$. Lastly, γ signifies the weight parameter.

## 3. Results

### 3.1. Experimental Platform

This section, which is structured with subheadings for clarity, succinctly details the implementation of GSCW-YOLO utilizing the PyTorch (2.0) deep learning framework. All training and testing activities were conducted on a system equipped with Ubuntu 18.04 operating system and an NVIDIA (Santa Clara, CA, USA) GeForce RTX 3090 graphics card with 24 GB of memory. Python served as the programming language for data preprocessing. The specifications of the hardware configuration and the hyperparameters adopted in this study are detailed in Table 3.

### 3.2. Evaluation Metrics

The performance of the model was assessed using several key metrics: precision, recall, mean Average Precision (mAP), model parameter size (in MB), and Frames Per Second (FPS). Precision is defined as the proportion of correctly identified positive instances out of all instances predicted as positive. Recall measures the fraction of correctly identified positive instances among all actual positive instances. The mAP metric offers a unified measure that balances precision and recall, providing an overarching view of the model’s effectiveness. Regarding resource usage, the model’s memory footprint is expressed in MB, and its processing speed, or the average number of frames processed per second, is denoted as FPS. The formulas for calculating these metrics are presented below.
(4)$$precision=\frac{1}{h}\sum_{i=1}^{h}\frac{{TP}_{i}}{{FP}_{i}+{TP}_{i}}$$
(5)$$recall=\frac{1}{h}\sum_{i=1}^{h}\frac{{TP}_{i}}{{FN}_{i}+{TP}_{i}}$$
(6)$$mAP=\frac{1}{h}\sum_{i=1}^{h}\int_{0}^{1}P_{i}(R_{i})dR_{i}$$
(7)$$FPS=\frac{1}{t_{pre}+t_{infer}+t_{post}}$$ where h represents the number of behavior categories, and i denotes the specific behavior category. ${TP}_{i}$ corresponds to correct classifications, where the model accurately recognizes the behavior represented by category i, while ${\;FP}_{i}$ indicates incorrect classification, suggesting the model identifies the behavior represented by category i erroneously. ${FN}_{i}$ corresponds to a classification omission where the actual behavior belongs to category i, but the model misidentifies it. The preprocessing time, inference time, and post-processing time are represented by $t_{pre}$, $t_{infer}$, and $t_{post}$, respectively.

### 3.3. Ablation Study on the Model’s Performance

The results of the ablation experiments are presented in Table 4. The integration of CARAFE resulted in a 1.2% increase in precision, 2% in recall, 1.4% in mean Average Precision (mAP), and a minor growth in parameter size to 6.3 MB. The exclusive addition of the Global Context Transformer (GCT) contributed to a 3.2% rise in precision, 0.8% in recall, and 1.3% in mAP, as it adeptly eliminated background noise, thus augmenting the model’s precision. Concurrently, the model’s size decreased to 5.9 MB. Incorporating only the SOD layer led to a 0.9% increase in precision, a substantial 3.9% in recall, and 1.2% in mAP, while reducing the model’s size to 5.9 MB. This improvement in recall is attributed to improved feature fusion, enabling more accurate target detection. Adding Wise-IoU independently enhanced precision by 1.3%, recall by 0.8%, and mAP by 0.9%, without altering the model’s parameters.

Additionally, the study investigated the synergistic effects of these modules relative to the baseline model. Adding the GCT module in conjunction with CARAFE for upsampling boosted precision by 2.4%, recall by 2.0%, and mAP by 1.4%, with the model size shrinking to 5.9 MB. A further enhancement with the SOD layer saw a 2.0% improvement in precision, 3.1% in recall, and 1.4% in mAP, keeping the model size virtually constant. The integration of all components—the Wise-IoU loss function, SOD, GCT, and CARAFE—led to a 3% increase in precision, 3.1% in recall, and a 2% rise in mAP while maintaining the model size at 5.9 MB. The compiled data from Table 4 and the detailed analysis above clearly demonstrate the significant impact of each introduced module on the algorithm’s efficacy.

### 3.4. Comparative Experiments Between Different Models

To assess the effectiveness of the proposed GSCW-YOLO model, it was benchmarked against six renowned models: YOLOv10n, YOLOv8n, YOLOv7 [36], YOLOv6n [37], YOLOv5n [38], CenterNet [39], and EfficientDet [20], using identical datasets and experimental settings. The results, presented in Table 5, reveal that the GSCW-YOLO model excels in precision and recall, achieving rates of 93.5% and 94.1%, respectively. In comparison, its nearest rivals, YOLOv8n and YOLOv5n, show a commendable balance between precision and recall. Despite CenterNet’s higher precision, its recall rate of 74.7% indicates possible detection misses. The highest mAP recorded was 97.5% for GSCW-YOLO, narrowly surpassing YOLOv8n and CenterNet. GSCW-YOLO is not only superior in performance but also more space-efficient, slimming down the YOLOv8n model by 0.3 MB to a compact size of 5.9 MB—a stark contrast to CenterNet’s 124.9 MB, reflecting a 21.2-fold size difference. Furthermore, GSCW-YOLO boasts the fastest recognition speed at 175 FPS, considerably faster than CenterNet’s 34 FPS. When compared to YOLOv10n, the most recent in the YOLO series, GSCW-YOLO demonstrates a 3.7% improvement in precision, a 6.2% increase in recall, and a 4.0% uplift in mAP, despite YOLOv10n’s smaller model size, which is smaller by a factor of approximately 5.2. Overall, our findings underscore GSCW-YOLO’s remarkable precision, efficiency in size, and speed.

### 3.5. Comparison of the Results of All Different Classes in the GoatABRD Dataset

Table 6 offers an analysis of various models’ recognition accuracy in identifying ten unique behaviors within GoatABRD, where GSCW-YOLO achieves superior mAP in nine of the ten categories. Notably, its performance on “limping” behavior is slightly below the top-performing CenterNet model by a mere 0.6 percentage points. CenterNet leads in recognizing “limping” with an mAP of 99.4%, closely followed by YOLOv8n, YOLOv10, EfficientDet, and GSCW-YOLO. GSCW-YOLO also outperforms other models in identifying “attacking” and “death” behaviors with mAPs of 97.3% and 98.3%, respectively, significantly outdistancing YOLOv7, which has the lowest performance in these categories. Moreover, GSCW-YOLO demonstrates excellent accuracy in detecting “gnawing” behavior with a leading mAP of 97.5%, while EfficientDet reports the lowest accuracy for the same behavior. This underscores GSCW-YOLO’s strong alignment with the targeted abnormal behaviors in this study.

Figure 4 underscores the experimental comparison between GSCW-YOLO and other models (YOLOv10n, YOLOv8n, YOLOv7, YOLOv5n, and CenterNet), highlighting GSCW-YOLO’s enhanced ability in recognizing small target behaviors through enlarged sections of selected images for better visualization.

Figure 4a illustrates the superior performance of GSCW-YOLO in recognizing the “drinking” behavior, in contrast to other models such as YOLOv8n, YOLOv7, YOLOv5n, and CenterNet, which failed in this respect. Noteworthy, GSCW-YOLO demonstrated a strong proficiency in small-object detection. Moreover, during the detection of the “gnawing” behavior, YOLOv7 experienced challenges with overlapping anchor boxes.

Figure 4b showcases the robust performance of GSCW-YOLO in complex scenarios. For instance, while other models struggled with anchor box overlap, leading to the misclassification of “gnawing” as “scratching”, GSCW-YOLO excelled by reducing background noise and capturing fine-grained action details for accurate recognition, even though YOLOv10n, avoiding anchor box overlap, misclassified behaviors and failed under occlusion conditions.

In Figure 4c, GSCW-YOLO’s precise classification capabilities are evident. When a goat near a water trough was not exhibiting the “drinking” behavior, only GSCW-YOLO correctly identified it as “standing”, while other models either misclassified or recognized it with low confidence. This exceptional accuracy and consistently high confidence highlight the model’s unmatched precision and reliability in behavior classification.

The adaptability and consistency of the GSCW-YOLO model under various lighting conditions are depicted in Figure 5a–d. The high detection accuracy in indoor and outdoor settings, as well as during nighttime, underscores the model’s effectiveness and reliability across different lighting environments, ensuring consistent and accurate behavior recognition.

## 4. Discussion

The effectiveness of livestock behavior recognition is crucially dependent on the quality and variety of the underlying datasets. However, existing datasets on dairy goat behaviors are limited in scope, primarily due to challenges in data collection, exhaustive annotation efforts, and the wide range of behaviors. This research performs a statistical analysis of methods, behavioral categories, and the efficiency of models in recent livestock behavior recognition studies, as detailed in Table 7. The analysis shows that traditional datasets focus mainly on common behaviors, such as standing, lying, and eating, and rarely include abnormal behaviors. Recognizing such behaviors is essential for continuous health monitoring and early detection of potential health issues or diseases. Current models, which predominantly target common behaviors, inadequately support decision-making or alert generation in practical livestock management due to their incapability in identifying abnormal behaviors. This emphasizes the need for comprehensive datasets covering a wide range of behaviors, including various abnormal ones. Accordingly, this study expands the representation of common behaviors and introduces a dataset that includes four abnormal behaviors (limping, attacking, death, and gnawing) and six common behaviors (standing, lying, eating, drinking, scratching, and grooming), thereby enhancing the diversity in dairy goat behavior recognition.

In 2016, Alvarenga et al. [40] analyzed 44 features from acceleration signals over three durations (3 s, 5 s, and 10 s) to classify behaviors, identifying the five most critical features for each period through random forest analysis and employing a decision tree algorithm for classification, achieving 92.5% accuracy. The advent of deep learning has reduced the reliance on manual feature engineering, enabling models, when properly trained on extensive datasets, to autonomously learn from raw data and improve performance significantly. Since 2020, deep learning techniques, such as convolutional neural networks, have become prevalent in behavior recognition. In 2022, Cheng’s team demonstrated this by training YOLOv5 on 9000 images for single sheep behavior recognition, reaching a mAP of 97.5% for identifying standing, lying[], feeding, and drinking behaviors [13]. As deep learning algorithms evolve, their application scope in behavior recognition has broadened from single livestock monitoring to comprehensive, multi-scale analysis, enhancing both the range of detectable behaviors and the overall accuracy and speed of recognition.

During the evaluation of the YOLOv8n model, it was observed that it occasionally produced overlapping anchor boxes for similar behavior types. This study proposed algorithmic refinements transforming YOLOv8n into GSCW-YOLO, aimed at reducing background noise and enhancing feature resolution and behavior differentiation. A comparative heatmap analysis, as depicted in Figure 6, conducted between YOLOv8n and GSCW-YOLO models, demonstrated that GSCW-YOLO effectively minimized irrelevant feature extraction, such as environmental elements, and excelled in capturing distinct goat features and their subtle variations. Moreover, a paired t-test revealed that the mAP values of GSCW-YOLO significantly differ from those of YOLOv8 across all 10 behavior categories, with a *p*-value of 0.00204. Given that this value is substantially below the standard significance threshold of 0.05, the difference in performance between the two models is considered statistically significant.

While this study has advanced behavior recognition by broadening the categories of behaviors and refining algorithms, it acknowledges limitations in employing abnormal behaviors for disease diagnosis. Future work will aim to incorporate expert knowledge in developing predictive models for diseases based on abnormal behaviors in dairy goats.

Although the GSCW-YOLO model is optimized specifically for dairy goat behavior recognition, its foundational principles, such as reducing interference from complex backgrounds and preserving fine-grained feature information, are broadly applicable. By customizing the training dataset to encompass behaviors and environmental contexts relevant to other livestock, such as cattle, sheep, or poultry, the model could be effectively adapted for their behavior recognition tasks. The real-time and lightweight nature of the GSCW-YOLO model makes it particularly suitable for integration into existing farm management systems. It can be deployed on edge devices or incorporated within smart agriculture platforms to facilitate continuous monitoring of livestock behaviors. This functionality allows farmers to gain insights that can improve animal welfare, boost productivity, and optimize farm operations.

However, substantial challenges remain in large-scale livestock farming settings, especially due to mutual occlusion among animals. Occlusion not only reduces the visibility of animals but also masks key behavioral features, which complicates the comprehensive capture of group dynamics within the dataset. While the model shows strong performance on smaller datasets with minimal occlusion, its efficacy in larger scenarios with 30–50 animals remains untested. Future research should focus on overcoming these limitations by using more extensive datasets and developing strategies to counteract the effects of occlusion, thereby ensuring the model’s robustness and scalability under diverse farming conditions.

## 5. Conclusions

Employing a non-contact method for precise, real-time monitoring of dairy goats’ behaviors—particularly for identifying abnormal behaviors—is essential for early detection of health concerns and environmental stress. This study introduces the GSCW-YOLO model, a lightweight, multi-scale approach for dairy goat behavior recognition that is based on YOLOv8n. Designed to mitigate background noise interference and effectively harness contextual information, it excels in recognizing behaviors from goats both distantly positioned from the camera and those exhibiting rapid behavioral shifts. To facilitate this, the GoatABRD dataset was developed, encompassing both common and abnormal behaviors—limping, attacking, death, gnawing, standing, lying, eating, drinking, scratching, and grooming. The GSCW-YOLO model is notable for its compact 5.9 MB size while achieving a precision rate of 93.5%, a recall rate of 94.1%, and a mean Average Precision (mAP) of 97.5%. When compared to the YOLOv8n model, the GSCW-YOLO model demonstrates enhancements in precision by 3.0 percentage points, recall by 3.1 percentage points, and mAP by 2.0 percentage points. Furthermore, GSCW-YOLO outstrips comparable methods, including YOLOv8n, YOLOv7, YOLOv6n, YOLOv5n, CenterNet, and EfficientDet, showcasing mAP improvements of 2.0, 7.9, 3.7, 3.0, 2.9, and 5.0 percentage points, respectively. The findings underscore the GSCW-YOLO model’s proficiency in discerning both fundamental and complex dairy goat behaviors, suggesting its adaptability for wider livestock behavior recognition applications such as swine, cattle, etc. This study marks a notable advancement in detecting diverse behavioral patterns amongst dairy goats at various scales, with the real-time recognition model poised to alert farmers to health and environmental conditions, ultimately promoting better welfare and farming practices.

## Figures and Tables

**Figure 1 animals-14-03667-f001:**
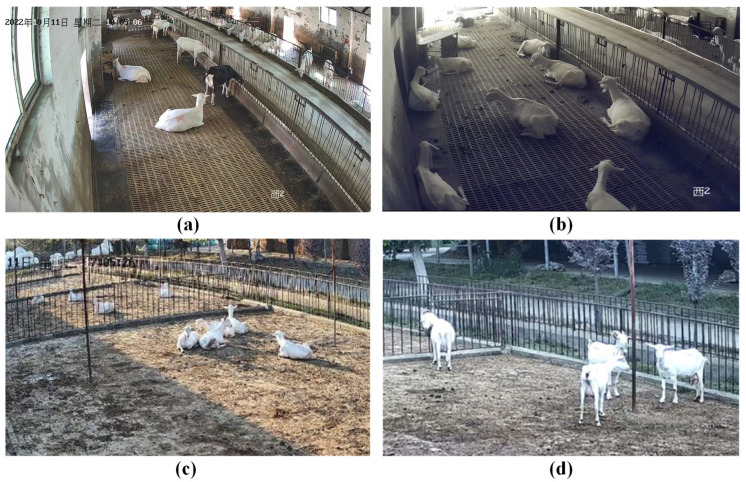
Examples of the recording of dairy goats in different scenes: (**a**) indoor recording; (**b**) indoor recording at night; (**c**) outdoor recording on a sunny day; and (**d**) outdoor recording on a cloudy day.

**Figure 2 animals-14-03667-f002:**
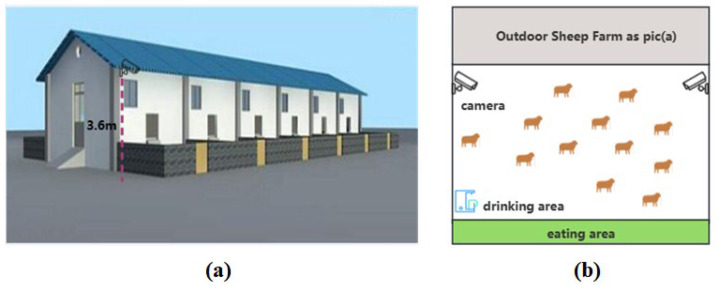
Dairy goat shed appearance and camera installation diagram. (**a**) The installation positions of the outdoor cameras. (**b**) The layout of the indoor cameras.

**Figure 3 animals-14-03667-f003:**
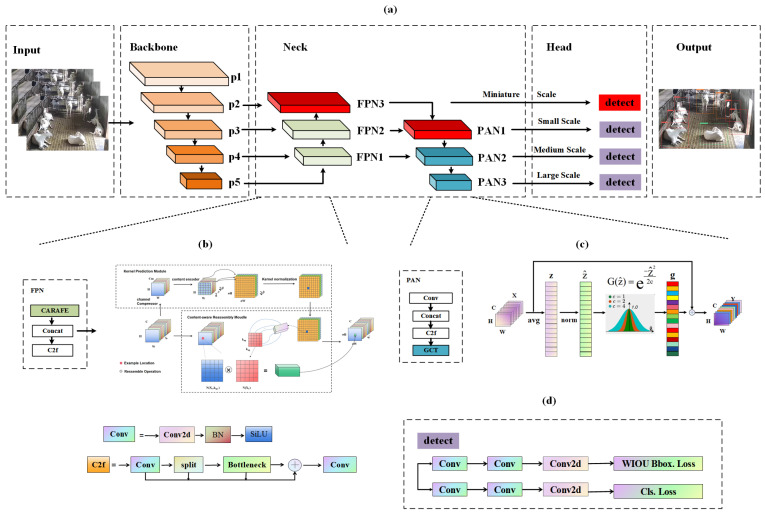
The architecture of GSCW-YOLO. (**a**) The overall architecture of GSCW-YOLO; (**b**) the structure of the CARAFE upsampling operator; (**c**) the specific structure of GCT; (**d**) the detailed structure of Wise-IoU.

**Figure 4 animals-14-03667-f004:**
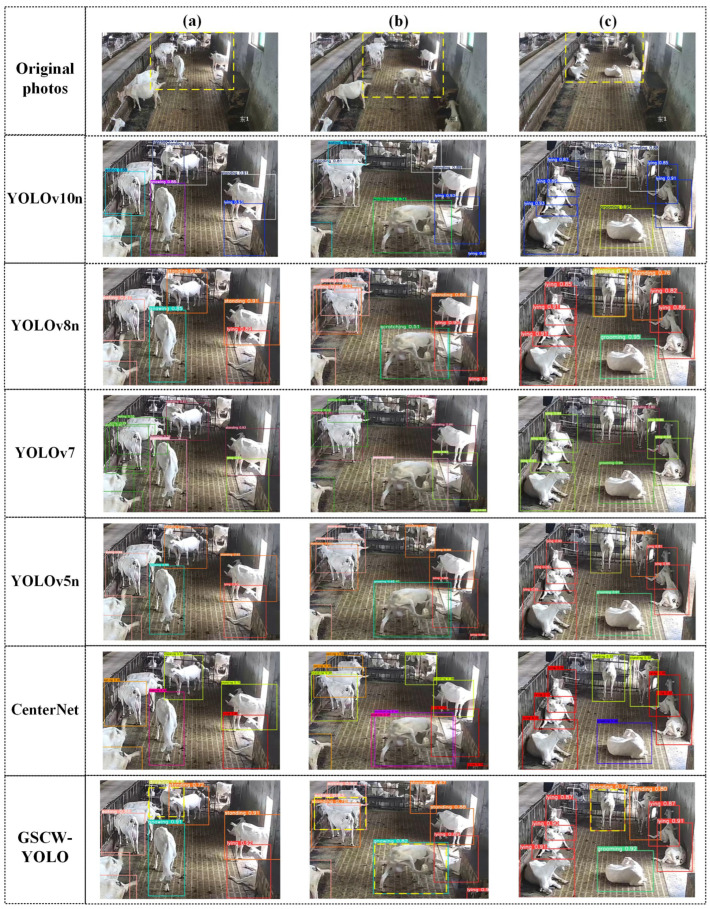
Comparative visualization results of GSCW-YOLO and other models. (**a**) Demonstrates the omission of the “drinking” behavior by YOLOv8n, YOLOv7, YOLOv5n, and CenterNet, with GSCW-YOLO accurately identifying it. (**b**) Highlights overlapping anchor boxes and misclassification of “gnawing” as “scratching” by YOLOv8n, YOLOv7, YOLOv5n, and CenterNet, while GSCW-YOLO achieves precise recognition. (**c**) Illustrates the accurate identification of “standing” by GSCW-YOLO in a challenging scenario, where YOLOv8n, YOLOv5n, YOLOv7, and CenterNet suffered from misclassification or low confidence.

**Figure 5 animals-14-03667-f005:**
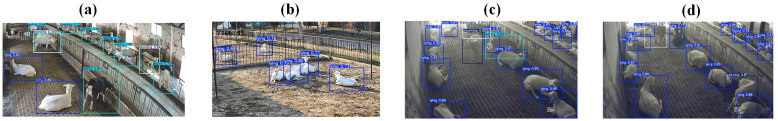
The performance of the GSCW-YOLO model under varying lighting conditions. (**a**) Highlights high detection accuracy in well-lit indoor environments. (**b**) Demonstrates robust behavior recognition, such as lying and standing, under outdoor sunlight. (**c**,**d**) Showcase effective detection of behaviors like lying, standing, and grooming during nighttime, emphasizing the model’s adaptability and reliability in varying lighting conditions.

**Figure 6 animals-14-03667-f006:**
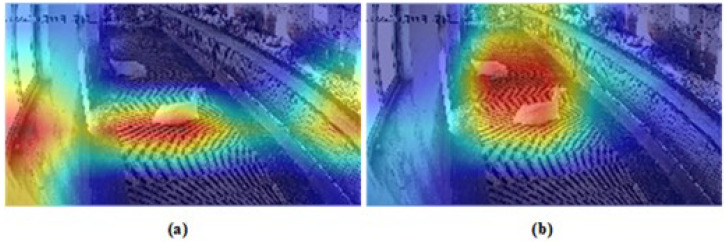
Comparative heatmap visualization of YOLOv8n and GSCW-YOLO. (**a**) The heatmap visualization results of YOLOv8n. (**b**) The heatmap visualization results of GSCW-YOLO.

**Table 1 animals-14-03667-t001:** Behavior recognition rules for dairy goats.

Typical Behaviors	Description	Instance
Standing	Goats maintain a stable quadrupedal posture, with their limbs either crossed or perpendicular to the ground.	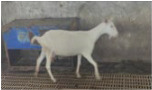
Lying	Goats lie flat on the ground with their legs folded under them or slightly extended.	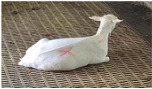
Eating	Goats chew food with their mouths in contact with the food, and their heads intersect with the feeding area.	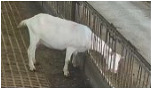
Drinking	Goats approach the water surface with their mouths, and their heads intersect with the water surface.	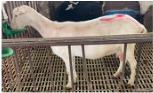
Scratching	Goats use their necks or bodies to rub against the walls, or they scratch their heads with their hooves.	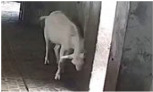
Grooming	Goats groom themselves by licking their abdominal region or other parts of their bodies with their tongues.	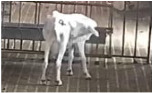
Limping	Goats exhibit an unsteady gait and often display symptoms of lameness or difficulty walking.	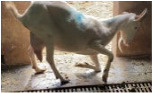
Attacking	Goats engage in head-butting by swiftly pushing their heads against the neck, head, or ears of another goat.	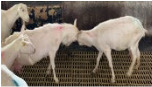
Gnawing	Dairy goats bite their hoof joints with their mouths.	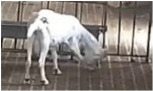
Death	Goats lie down on the ground horizontally, remaining still and unresponsive.	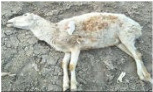

**Table 2 animals-14-03667-t002:** The behavior instances distribution of the training, validation, and testing sets.

Behavior	Train_set	Val_set	Test_set	Total
Standing	8274	1931	1113	11,318
Lying	9033	2319	1291	12,643
Eating	8161	1995	1058	11,214
Drinking	897	223	126	1246
Scratching	866	220	125	1211
Grooming	704	188	112	1004
Limping	548	144	73	765
Attacking	411	103	55	569
Gnawing	353	88	63	504
Death	457	103	54	614

**Table 3 animals-14-03667-t003:** Hardware and hyperparameter information.

Configuration Item	Value
CPU	Intel(R) Xeon(R) CPU E5-2683 v3 @ 2.00 GHz
GPU	NVIDIA GeForce RTX 3090
Operating system	Ubuntu 18.04.6 LTS
Learning rate	0.01
Training epochs	150
Batch size	32
Image size	224 × 224
Optimizer	SGD

**Table 4 animals-14-03667-t004:** Results of the ablation experiment based on GSCW-YOLO.

Model	Precision (%)	Recall (%)	mAP (%)	MB
YOLOv8n	90.5	91.0	95.5	6.2
YOLOv8n+CARAFE	91.7	93.0	96.9	6.3
YOLOv8n+GCT	93.7	91.8	96.8	5.9
YOLOv8n+SOD	91.4	94.9	96.7	6.1
YOLOv8n+Wise-IOU	92.8	91.8	96.4	6.2
YOLOv8n+GCT+CARAFE	92.9	93.0	96.9	5.9
YOLOv8n+GCT+CARAFE+SOD	92.5	94.1	97.3	5.9
GSCW-YOLO	93.5	94.1	97.5	5.9

**Table 5 animals-14-03667-t005:** Comparison of the results of different models.

Model	Percentage (%)			MB	FPS
	Precision	Recall	mAP		
YOLOv10n	89.8	87.9	93.5	5.7	142
YOLOv8n	90.5	91.0	95.5	6.2	126
YOLOv7	83.3	85.7	89.6	74.8	169
YOLOv6n	91.7	86.3	93.8	32.7	161
YOLOv5n	91.0	88.9	94.5	19.6	125
CenterNet	93.2	74.7	94.6	124.9	34
EfficientDet	88.8	89.4	92.5	15.1	113
GSCW-YOLO	93.5	94.1	97.5	5.9	175

**Table 6 animals-14-03667-t006:** Comparison of the results of different behavior categories in the GoatABRD dataset.

Model	Standing	Lying	Eating	Drinking	Scratching	Grooming	Limping	Attacking	Death	Gnawing
YOLOv10n	91.7	98.5	96.0	93.6	97.8	98.0	98.9	92.3	74.4	94.1
YOLOv8n	91.9	98.0	94.6	95.0	97.2	98.4	99.0	93.8	96.3	95.8
YOLOv7	87.8	95.9	91.6	92.1	95.8	90.9	96.9	89.6	62.4	93.1
YOLOv6n	90.3	96.9	92.0	93.2	96.8	97.2	97.8	92.7	86.5	95.0
YOLOv5n	91.0	97.7	93.7	93.1	97.5	97.5	98.7	96.7	82.7	96.2
CenterNet	94.4	98.7	95.6	91.8	95.2	94.2	99.4	91.1	92.2	93.1
EfficientDet	92.3	98.3	96.3	90.3	93.9	95.1	98.9	89.7	77.7	92.2
GSCW-YOLO	94.5	99.2	97.2	95.9	98.1	98.7	98.8	97.3	98.3	97.5

**Table 7 animals-14-03667-t007:** Recent Statistics of Behavior Categories in Livestock Behavior Recognition Studies.

Livestock	Methods	Behavior Categories	Performance (%)		
			mAP	Accuracy	F1
Sheep (Alvarenga et al., 2016) [40]	Decision Tree Algorithm	grazing, lying, running, standing, walking		92.5	
Sheep (Decandia et al., 2018) [41]	canonical discriminant analysis (CDA), and discriminant analysis (DA)	Grazing, ruminating		89.7	
Pig (Nasirahmadi et al., 2019) [42]	Support Vector Machine (SVM)	different pig lying postures		94.4	
Single Cow (Yin et al., 2020) [20]	EfficientNet-LSTM	lying, standing, walking, drinking, feeding	97.8		
Pig (Yang et al., 2021) [43]	Faster R-CNN	mounting	95.2		
Sheep (Cheng et al., 2022) [13]	YOLOv5	standing, lying, feeding, drinking	97.4		
Cows (Lodkaew et al., 2023) [44]	CowXNet	estrus			89.0
Cows (Wang et al., 2024) [45]	Improved YOLOv8n	estrus, mounting			93.7

## Data Availability

The data and code generated and analyzed in this study will be available at https://github.com/sunianbei/GoatABRD-Dataset (accessed on 21 November 2024) in the future upon request.

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
