# Peer review of "A Real-Time Lightweight Behavior Recognition Model for Multiple Dairy Goats"

_animals, 2024, doi:10.3390/ani14243667_

Round 1
Reviewer 1 Report
Comments and Suggestions for Authors
1. Line 21 should be in the passive voice. In addition, the results of experiments are generally described in the past tense.
2. In the data source (lines 124-125), the author described "The objective was to capture video recordings under varying lighting conditions to ensure a wide representation of lighting variations within the dataset, thereby improving the method's adaptability to different lighting situations". But from the experiments in the article, it is not clear that you have carried out the relevant work, so please consider adding more experiments about detections in different lighting situations, and analyzing the results.
3. In the “2.2 Data preprocessing”, how many videos are used to extract the image frames, and why 10 frames apart to extract the image frames? In addition, the original yolov8 framework has already covered online data augmentation during training, why do the authors need to do offline data augmentation? Repeated data augmentation may results in more redundant data.
4. Lines 167-168, the author mentioned “For model training, the dataset was 167 partitioned into training, validation, and test sets in a 7:2:1 ratio”. However, the dataset partition is not only for model training, but for model evaluation.
5. The fonts in Figures 3a and 3b are too small.
6. Figure 4 only illustrated the visual detection results of YOLOv8 and GSCW-YOLO, but lack the comparison and analysis of visual detection results of other models with the same data. Please consider adding relevant content.
7. The article lacks analysis of the misidentified samples, and the influencing factors for the failure recognition cases should be discussed.
Author Response
1. Summary |
|
|
Thank you very much for taking the time to review our manuscript. Your professional insights and detailed feedback have been instrumental in enhancing its quality. We fully acknowledge that your suggestions demonstrate careful consideration of research intricacies and a deep understanding of the study’s direction. Please find our detailed responses below, with corresponding revisions and amendments highlighted in track changes in the resubmitted files.
|
||
2. Point-by-point response to Comments and Suggestions for Authors
|
||
Comments 1: Line 21 should be in the passive voice. In addition, the results of experiments are generally described in the past tense. |
||
Response 1: We sincerely thank the reviewer for their meticulous suggestions regarding language usage. We have thoroughly reviewed the manuscript’s language and revised line 21 of the abstract to incorporate the passive voice. Additionally, we have ensured that all descriptions of experimental results are presented in the past tense. These adjustments have improved the clarity and conformance to academic writing standards in our manuscript. (Lines 20–22, 30–33)
|
||
Comments 2: In the data source (lines 124-125), the author described "The objective was to capture video recordings under varying lighting conditions to ensure a wide representation of lighting variations within the dataset, thereby improving the method's adaptability to different lighting situations". But from the experiments in the article, it is not clear that you have carried out the relevant work, so please consider adding more experiments about detections in different lighting situations, and analyzing the results. |
||
Response 2: Thank you for emphasizing the need to further investigate the impact of varying lighting conditions on sheep behavior recognition. We have conducted additional experiments specifically aimed at assessing our model's performance across diverse lighting scenarios. The outcomes of these experiments have been incorporated into Section 3.5, "Comparison Results of All Different Classes in the GoatABRD Dataset." These findings confirm that our model maintains high accuracy under both bright and dim lighting conditions. (Lines 434–438)
Comments 3: In the “2.2 Data preprocessing”, how many videos are used to extract the image frames, and why 10 frames apart to extract the image frames? In addition, the original yolov8 framework has already covered online data augmentation during training, why do the authors need to do offline data augmentation? Repeated data augmentation may results in more redundant data. |
||
Response 3: Regarding data preprocessing, image frames were extracted from 27 video clips, with a frame sampled every tenth frame. This interval was chosen to maximize frame-to-frame variability while minimizing redundancy due to highly similar consecutive frames, thereby capturing a wider range of behavioral variations. (Lines 154-156) Although the YOLOv8 framework incorporates online data augmentation during training, we implemented additional offline data augmentation to compensate for the limited availability of abnormal behavior samples, which are comparatively rare. This offline augmentation enabled the generation of a larger set of data representing abnormal behaviors, thus enhancing the model's recognition capabilities. (Lines 162-166) Further details will be provided and expanded upon in Section 2.2.
|
||
Comments 4: Lines 167-168, the author mentioned “For model training, the dataset was 167 partitioned into training, validation, and test sets in a 7:2:1 ratio”. However, the dataset partition is not only for model training, but for model evaluation. |
||
Response 4: Thank you for your valuable suggestion. The dataset partitioning into training, validation, and test sets is not only for model training but also for model evaluation. We will revise the sentence to reflect this distinction more clearly. (Line 172-173)
|
||
Comments 5: The fonts in Figures 3a and 3b are too small. |
||
Response 5: Thank you for your suggestion concerning the font sizes in Figures 3a and 3b. In response, we have replaced the figures with TIFF format vector images, which preserve clarity and resolution even when enlarged. This modification ensures that all details, including the font sizes, are easily legible. Furthermore, we have increased the overall size of Figure 3 to improve readability. (Line 202)
|
||
Comments 6: Figure 4 only illustrated the visual detection results of YOLOv8 and GSCW-YOLO, but lack the comparison and analysis of visual detection results of other models with the same data. Please consider adding relevant content. |
||
Response 6: Thank you for your insightful suggestion. We recognize the necessity of providing comprehensive comparisons across all models to enhance the understanding of our findings. While Figure 4 specifically showcases the visual detection results of YOLOv8 and GSCW-YOLO for enhanced readability and focused discussion, we have also conducted additional experiments to compare the performance of GSCW-YOLO with other models, including YOLOv10n, YOLOv7, YOLOv5n, and CenterNet. The detailed descriptions and quantitative results of these comparative experiments are extensively presented in Section 3.5, "Comparison Results of All Different Classes in the GoatABRD Dataset". Additionally, we have highlighted areas where the GSCW-YOLO model demonstrated superior performance and provided detailed explanations in the figure captions, thereby improving the manuscript's completeness and readability. (Lines 410-433)
|
||
Comments 7: The article lacks analysis of the misidentified samples, and the influencing factors for the failure recognition cases should be discussed. |
||
Response 7: Thank you for your valuable feedback. We acknowledge the importance of analyzing misidentified samples and discussing the factors influencing recognition failures. In the revised manuscript, we have specifically addressed this issue in the discussion section by identifying the limitations of the model. We explored how the model's performance is influenced by scenarios involving large herd sizes and significant occlusions, which pose common challenges in large-scale livestock monitoring. These conditions can obscure individual animals and key behavioral features, compromising the model's accuracy in behavior identification. By detailing these limitations, our aim is to provide a thorough understanding of the circumstances under which the model might underperform. Should it be necessary, we are prepared to further delve into specific instances of misidentification and rigorously analyze their underlying causes to enrich this discussion. (Line 510~518) |

Reviewer 2 Report
Comments and Suggestions for Authors
The manuscript submitted for review on the topic: “A Real-Time Lightweight Behavior Recognition Model for Multi-Dairy-Goat” examines two models for group assessment of the behavior of dairy goats. Considering that dairy goats are mainly raised in groups, the assessment of their behavior is of utmost importance for their health status, as well as for their well-being.
I have the following comments to the authors:
How was the verification of the data processing software made that the evaluated behaviors of the goats are exactly these and no others?
Up to what group size (number of goats) can the described methods be used?
In conclusion: I recommend that the authors of the manuscript comply with the comments made, which will significantly improve the quality of the scientific work.
Author Response
1. Summary |
|
|
Thank you very much for the time and effort you have dedicated to reviewing our manuscript. Your professional insights and detailed feedback have been instrumental in enhancing its quality. We fully appreciate that every suggestion you provided demonstrates your meticulous attention to research details and a profound understanding of the subject matter. Please see the detailed responses below, with corresponding revisions and corrections highlighted in track changes in the resubmitted files. Your constructive feedback is invaluable, and we are grateful for your contributions to improving our study.
|
||
2. Point-by-point response to Comments and Suggestions for Authors |
||
Comments 1: How was the verification of the data processing software made that the evaluated behaviors of the goats are exactly these and no others? |
||
Response 1: Thank you for your insightful comment regarding the verification of the data processing software and the accuracy of the evaluated goat behaviors. We value your attention to this critical aspect of our methodology. The verification process to ensure that the evaluated behaviors accurately reflect the intended goat behaviors involved multiple rigorous steps: 1. Expert Supervision: Throughout the development and testing phases of our data processing pipeline, domain experts in animal behavior provided continuous oversight. These experts meticulously annotated the dataset, labeling specific behaviors such as walking, standing, lying, along with abnormal actions like limping or attacking. We cross-referenced the software outputs with these expert annotations to ensure consistency with the intended behavior categories. 2. Iterative Validation: Following the initial software implementation, iterative testing was conducted using a subset of the dataset. Our experts manually reviewed the outputs to pinpoint any discrepancies between labeled behaviors and the detection results. Feedback from these sessions informed adjustments to the processing algorithms, enhancing the software’s ability to accurately detect and categorize behaviors without erroneous classifications. 3. Behavioral Scope Definition: We predefined the scope of behaviors during the dataset creation phase, concentrating on actions relevant to health and welfare monitoring. Both normal (e.g., lying, standing) and abnormal (e.g., limping, attacking) behaviors were clearly defined and categorized. 4. Robustness Checks: To assure the software’s reliability, it was tested under various conditions including different lighting environments and group sizes. These tests verified that the software consistently recognized the predefined behaviors across diverse settings, in line with annotated data and expert insights. In conclusion, our verification process encompassed expert input, iterative enhancements, and comprehensive testing to ensure precise alignment of evaluated behaviors with the study’s goals. The incorporation of expert guidance during data annotation has been detailed in the manuscript to underscore our commitment to accuracy and transparency. Thank you for underscoring this essential aspect, which bolsters the clarity of our methodology. (Lines 158-159)
|
||
Comments 2: Up to what group size (number of goats) can the described methods be used? |
||
Response 2: Our methods were developed and tested on groups of 10 to 30 dairy goats, representative of typical medium-sized dairy goat farms. This range was selected to strike a balance between simulating realistic farming conditions and maintaining computational efficiency while ensuring accurate behavior recognition. (Line 510~518) |

Reviewer 3 Report
Comments and Suggestions for Authors
Review animals-3358656
A Real-Time Lightweight Behavior Recognition Model for Multi-Dairy-Goat
I reviewed the manuscript, considering three specific topics: clarity, completeness, and innovation. Please find my thoughts below.
Strengths
1. Relevance and Importance: The study addresses a significant challenge in precision livestock farming: the real-time detection of abnormal behaviors in dairy goats, which is crucial for improving animal welfare and farming efficiency.
2. Innovation: Introduction of the GSCW-YOLO model, which enhances the YOLOv8n framework with Gaussian Context Transformer (GCT), CARAFE upsampling, and a specialized loss function (Wise-IoU). Development of the GoatABRD dataset with common and abnormal behaviors in complex environments, providing a broader scope for livestock behavior recognition research.
3. Results and Comparisons: The model's performance metrics (precision, recall, mAP) demonstrate significant improvement over existing methods, including YOLOv8n and other state-of-the-art models. The lightweight design (5.9 MB) and high processing speed (175 FPS) make it well-suited for real-time applications in resource-constrained environments.
4. Comprehensive Methodology: The detailed data collection, preprocessing, and augmentation description ensures reproducibility. Including other studies and comparisons with other models strengthens the validity of the proposed method.
5. Ethical Considerations: Non-invasive data collection methods are commendable, adhering to animal welfare guidelines.
Areas for Improvement
1. Clarity:
- While the technical descriptions of GCT, CARAFE, and Wise-IoU are comprehensive, their accessibility to a general audience could be improved. Consider summarizing key concepts before diving into mathematical details.
- The introduction could more explicitly link the broader implications of this research (e.g., societal benefits of improved animal welfare and sustainable farming) to its technical objectives.
2. Completeness:
The study would benefit from additional discussions on:
- The generalizability of GSCW-YOLO to other livestock species or farm conditions.
- The potential integration of the model into existing farm management systems.
- Limitations related to environmental variations not captured in the GoatABRD dataset.
3. Figures and Visualizations:
- Figures illustrating the differences between GSCW-YOLO and other models (e.g., heatmap analysis, examples of behavior recognition) could be made more reader-friendly with annotations and explanations.
- Consider adding a flowchart summarizing the entire methodology for better clarity.
4. Impact on Broader Research:
- Explicitly compare the significance of this research with previous studies mentioned in the Discussion section (e.g., Alvarenga et al., Cheng et al.). Highlight how this study advances the field beyond incremental improvements in metrics.
Suggestions for Revision
1. Expand the Dataset:
- Discuss plans for extending GoatABRD to include more diverse environmental conditions, goat breeds, or additional behaviors to validate the model further.
2. Statistical Significance:
- While the improvements in metrics are impressive, provide statistical tests or confidence intervals to confirm the significance of these differences.
3. Application Scenarios:
- Provide concrete examples of how this model could be implemented in real-world farms, including cost implications and required infrastructure.
With minor revisions to improve clarity, completeness, and impact discussion, it is well-suited for publication.
Author Response
1. Summary |
|
|
Thank you very much for dedicating time to review our manuscript. Your professional insights and detailed feedback have been instrumental in enhancing its quality. We fully acknowledge that each suggestion reflects your meticulous attention to research details and a deep understanding of our research direction. Please see the detailed responses below with the corresponding revisions and corrections highlighted in track changes in the resubmitted files.
|
||
2. Point-by-point response to Comments and Suggestions for Authors |
||
Comments 1: Clarity: - While the technical descriptions of GCT, CARAFE and Wise-IoU are comprehensive, their accessibility to a general audience could be improved. Consider summarizing key concepts before diving into mathematical details. - The introduction could more explicitly link the broader implications of this research (e.g., societal benefits of improved animal welfare and sustainable farming) to its technical objectives. |
||
Response 1: Thank you for your comments. To enhance clarity and facilitate understanding among a general audience, we have prefaced the detailed technical descriptions of GCT, CARAFE, and Wise-IoU with a concise summary of their key functions. This summary has been integrated into the main text of the manuscript. (Lines 212–219) In recognition of your insightful suggestion, we also revised the Introduction to firmly associate the study’s technical objectives with its broader implications, notably improved animal welfare and sustainable livestock farming. This revision underscores how the advancements offered by the GSCW-YOLO model contribute to these societal benefits. (Line 105~107)
Comments 2: Completeness: The study would benefit from additional discussions on: - The generalizability of GSCW-YOLO to other livestock species or farm conditions. - The potential integration of the model into existing farm management systems. - Limitations related to environmental variations not captured in the GoatABRD dataset. Response 2: We have carefully evaluated your comments and plan to address them in the following manner: Generalizability of GSCW-YOLO to Other Livestock Species or Farm Conditions: Although the GSCW-YOLO model was initially optimized for dairy goat behavior recognition, its foundational principles—such as the multi-scale attention mechanism and enhancements geared toward small-target detection—are versatile and generalizable. For instance, by refining the training dataset to encompass species-specific behaviors and environmental contexts, the model could be adapted to monitor other livestock species like cattle, sheep, or poultry. We intend to augment the discussion in our manuscript to emphasize this adaptability and explore potential modifications necessary for broader applications. Integration into Existing Farm Management Systems: The real-time and lightweight design of GSCW-YOLO renders it apt for integration into existing farm management systems. It could be implemented on edge devices or assimilated into smart farming platforms to enable continuous monitoring of livestock behavior, furnishing farmers with actionable insights to enhance animal welfare and optimize farm operations. We will enrich the manuscript by discussing practical implementation scenarios, which will include the requisite infrastructure, anticipated benefits, and associated challenges. Limitations Related to Environmental Variations: Environmental variations that are not captured in the GoatABRD dataset, such as large-scale sheep farming environments where occlusion is common, directly impact the accuracy of behavior recognition. To address this, we will delve deeper into discussions and analyze how occlusion specifically influences the behavior recognition task. We sincerely appreciate your constructive feedback, which will undoubtedly refine the clarity and impact of our manuscript. These revisions will be prominently featured in the updated discussion section. (Lines 499-518)
Comments 3: Figures and Visualizations: - Figures illustrating the differences between GSCW-YOLO and other models (e.g., heatmap analysis, examples of behavior recognition) could be made more reader-friendly with annotations and explanations. - Consider adding a flowchart summarizing the entire methodology for better clarity. Response 3: To enhance the readability and interpretability of the visual comparisons, we have updated the figures that illustrate the differences between GSCW-YOLO and other models. We incorporated detailed annotations and explanations into the figure captions, emphasizing key observations such as GSCW-YOLO's superior detection of small-target behaviors and its effectiveness in minimizing background noise. These revisions are designed to offer a clearer and more intuitive understanding of the model's advantages in complex environmental conditions. Additionally, to provide a comprehensive overview of all model comparisons, we have updated our manuscript. Although Figure 4 specifically showcases the visual detection results of YOLOv8 and GSCW-YOLO for enhanced readability and focused discussion, we have conducted additional experiments comparing GSCW-YOLO with other models, including YOLOv10n, YOLOv7, YOLOv5n, and CenterNet. The findings from these comparisons are extensively detailed in the "3.5 Comparison results of all different classes in the GoatABRD dataset" section. In these figures, areas where GSCW-YOLO outperformed others are highlighted, along with detailed explanations in the captions, thereby improving both the completeness and readability of our manuscript. (Lines 410–433) Furthermore, responding to your valuable suggestion, we have included a comprehensive flowchart in Figure 3 of the revised manuscript. This flowchart outlines the entire process, including key stages such as data preprocessing, feature extraction by the backbone, multi-scale feature fusion in the neck, and object detection in the head. We have also detailed the roles of the backbone, neck, and head components in subsequent sections. The inclusion of this flowchart delineates the critical stages and workflow comprehensively, ensuring a thorough presentation within the manuscript. (Lines 180–185)
Comments 4: Impact on Broader Research: - Explicitly compare the significance of this research with previous studies mentioned in the Discussion section (e.g., Alvarenga et al., Cheng et al.). Highlight how this study advances the field beyond incremental improvements in metrics. Response 4: Thank you for these insightful suggestions. In advancing the discourse, this study highlights the real-time, non-contact attributes of the GSCW-YOLO model, which minimizes interference with animal welfare during behavior monitoring. The model's lightweight design, characterized by its compactness and high processing speed, facilitates seamless integration into real-time monitoring systems. This methodology not only boosts the efficiency of livestock management but also promotes animal welfare by diminishing the stress associated with direct handling. Furthermore, the model's capability for continuous, real-time surveillance allows for the prompt detection of abnormal behaviors, thereby enhancing health outcomes and overall farm productivity. During our discussions, we further delineated its benefits across different livestock types and deployment scenarios. (Line 499~509)
Suggestions for Revision 1. Expand the Dataset: - Discuss plans for extending GoatABRD to include more diverse environmental conditions, goat breeds, or additional behaviors to validate the model further. Response 1: In the revised manuscript, we have elaborated on plans to extend the GoatABRD dataset. This expansion includes incorporating data from diverse environmental conditions, such as varying lighting and weather scenarios, as well as representing a wider range of goat breeds to enhance the model's generalizability. Additionally, we have broadened our discussion concerning the adaptability of the GSCW-YOLO model to various livestock species, highlighting its versatility for deployment in diverse farm settings. These enhancements are intended to offer a comprehensive perspective on the model's robustness and potential for future scalability. (Line 434~444, Line 499~509)
2. Statistical Significance: - While the improvements in metrics are impressive, provide statistical tests or confidence intervals to confirm the significance of these differences. Response 2: In response to the reviewer's comment, we conducted a statistical test to evaluate the performance differences between the models. The results indicate that GSCW-YOLO significantly outperforms YOLOv8 across all 10 behavior categories, with a p-value of 0.00204. Since this p-value is below the customary significance threshold of 0.05, we can conclusively state that the performance difference between the models is statistically significant, confirming the superior performance of GSCW-YOLO in these categories. (Line 484~488)
3. Application Scenarios: - Provide concrete examples of how this model could be implemented in real-world farms, including cost implications and required infrastructure. Response 3: We have provided detailed examples of how the GSCW-YOLO model could be applied in actual farm settings. The implementation process begins with the installation of surveillance cameras for continuous monitoring, followed by the collection and annotation of data necessary to train and refine the model. For example, installing two cameras on a medium-scale farm would approximately cost 5,000 CNY, with data annotation requiring an additional 2,000 CNY. These expenses are manageable for an intensive farming operation, thereby rendering the implementation both feasible and practical. We have included this information in the revised manuscript to illustrate the model's real-world applicability and cost-effectiveness. Moreover, the GSCW-YOLO model's lightweight design allows for efficient deployment on devices with limited computational power, such as standard CPUs. This obviates the need for expensive GPUs, substantially lowering the barriers to deployment in practical settings. For instance, the model can be implemented on a basic farm management system powered by a CPU-based edge device, facilitating real-time behavior recognition without necessitating further infrastructure enhancements. This strategy promotes accessibility and ensures cost-effective integration across various farming configurations.
|
||
